# Tract-specific analysis improves sensitivity of spinal cord diffusion MRI to cross-sectional and longitudinal changes in amyotrophic lateral sclerosis

Pramod Kumar Pisharady [1,2✉], Lynn E. Eberly [1,3], Ian Cheong [1], Georgios Manousakis[4], Gaurav Guliani [4], H. Brent Clark[5], Mark Bathe [2], David Walk [4,6] & Christophe Lenglet [1,6]

Amyotrophic lateral sclerosis (ALS) is a late-onset fatal neurodegenerative disease that causes progressive degeneration of motor neurons in the brain and the spinal cord. Corticospinal tract degeneration is a defining feature of ALS. However, there have been very few longitudinal, controlled studies assessing diffusion MRI (dMRI) metrics in different fiber tracts along the spinal cord in general or the corticospinal tract in particular. Here we demonstrate that a tract-specific analysis, with segmentation of ascending and descending tracts in the spinal cord white matter, substantially increases the sensitivity of dMRI to disease-related changes in ALS. Our work also identifies the tracts and spinal levels affected in ALS, supporting electrophysiologic and pathologic evidence of involvement of sensory pathways in ALS. We note changes in diffusion metrics and cord cross-sectional area, with enhanced sensitivity to disease effects through a multimodal analysis, and with strong correlations between these metrics and spinal components of ALSFRS-R.

[1] Center for Magnetic Resonance Research, Department of Radiology, University of Minnesota, Minneapolis, MN 55455, USA. [2] Department of Biological Engineering, Massachusetts Institute of Technology, Cambridge, MA 02139, USA. [3] Division of Biostatistics, School of Public Health, University of Minnesota, Minneapolis, MN 55455, USA. [4] Department of Neurology, University of Minnesota, Minneapolis, MN 55455, USA. [5] Department of Laboratory Medicine and Pathology, University of Minnesota, Minneapolis, MN 55455, USA. [6] These authors jointly supervised this work: David Walk, Christophe Lenglet. ✉email: pramodkp@umn.edu

Amyotrophic lateral sclerosis (ALS) is a fatal neurodegenerative disorder characterized by progressive weakness of the limb, bulbar, and respiratory muscles[1]. Accurate modeling and quantification of alterations in the central nervous system in vivo has the potential to improve our understanding of ALS and other neurodegenerative disorders. Although the pathogenesis of ALS is still uncertain, the cardinal clinical and pathologic features are supported by ample neuroimaging evidence of degeneration of motor and extramotor neural pathways[2]. These insights have been achieved, in part, by using diffusion tensor imaging (DTI)[3–5] in combination with tractography methods[6].

DTI studies in ALS have shown that the degree of directionality and organization of diffusion, measured by fractional anisotropy (FA), is useful for assessing degeneration in tissue structure (see reviews[3–5]). Most of these studies reported findings in brain regions in ALS. There are fewer studies of the corticospinal tract (CST) in the spinal cord, despite its prominent pathologic involvement in ALS; indeed, the disease is named in part for the pathological findings in the lateral CST. The simpler anatomy of the spine compared with the brain, with limited fiber crossing structures, makes DTI well suited for spinal analysis, with the potential to lead to more reliable and reproducible biomarkers.

Cross-sectional differences in spinal cord DTI metrics have been reported and include lower FA and higher radial diffusivity (RD) and mean diffusivity (MD)[7,8]. In a longitudinal study of ALS patients and healthy controls utilizing a 1.5 T scanner, Agosta et al.[9] reported decreased FA, increased MD, and decreased cord cross-sectional area (CSA) in whole cervical cord in ALS patients over time. These imaging changes however did not correlate with functional decline, although their mean follow-up was relatively brief (9 months). Mendili et al.[10] subsequently reported longitudinal findings utilizing a 3 T scanner but without a healthy control cohort. In their study, FA is extracted from manual segmentation of the CST through the entire C2-T2 region. They found significant correlations between FA and the ALS Functional Rating Scale-Revised (ALSFRS-R) leg sub-score and between cord CSA and the ALSFRS-R arm sub-score at baseline and follow-up scans. They postulated that the reduction in cervical cord FA reflects degeneration of CST axons to the lower limbs, whereas reduction in cord CSA is more reflective of loss of anterior horn cells to the upper limbs. A recent cross-sectional study of 10 ALS and 20 control participants by Rasoanandrianina et al.[11] recapitulated evidence of decreased CSA and reduced CST and white matter FA. Diffusion metrics were reported at the C2 and C5 levels only. Functional status correlated with CSA but significant correlations between functional scores and DTI metrics were not reported. The study included segmentation of spinal cord gray matter and inhomogeneous magnetization transfer. Among several limitations of their study, the authors cited a slice thickness of 10 mm, a small cohort, and the lack of longitudinal data. Another very recent cross-sectional study with 14 ALS and 15 control participants by Patzig et al.[12] reported a reduction in FA in whole cross-section at the C2–C4 level and the T1–T3 level. They also reported a trend towards a statistically significant FA reduction in the anterolateral regions at the C5–C7 level, but not in the corresponding posterior regions. The study was limited to the analysis of FA only and did not include longitudinal data.

In summary, spinal cord imaging in ALS offers the potential to devise non-invasive in vivo measures of the clinical and pathologic hallmark of ALS, the degeneration of the CST and anterior horns. Prior studies of cervical cord in ALS have consistently demonstrated reductions in cord CSA and FA, but with inconsistent clinical correlations. Prior studies have varied considerably in many features, including field strength, cord segments analyzed, extent of cord segmentation, the presence or absence of a healthy control cohort, and the presence or absence of longitudinal data.

To confirm and extend previous findings while addressing some of the limitations of prior studies, we performed a longitudinal study of spinal cord CSA and diffusion at high field strength with analysis of individual spinal pathways and an along-the-tract analysis, using tractography, to evaluate the continuous variation of diffusion metrics along the cervical spine from C2 to C6 and to identify the spinal levels most affected in ALS. We evaluated participants with ALS and a cohort of healthy control participants, with group matching for age and gender, over 1 year. We used a varying coefficient model (VCM) to perform a multimodal statistical analysis of imaging metrics such as FA and CSA along the tracts, and to account for the effects of covariates such as age and gender[13]. We performed tract-specific analysis using a semi-automated method to segment the spinal cord into gray/white matter and into specific ascending/descending tracts. We noted significant difference in FA in the lateral CST and the spinal lemniscus. We also observed significant differences in CSA of the cord and enhanced sensitivity to disease effects via a multimodal analysis combining FA and CSA. FA and CSA correlated with the spinal components of the ALSFRS-R functional score. Longitudinal analysis demonstrated significant increases in RD and MD in the descending tracts, in the lateral CST in particular.

## Results

**Cohort characteristics**. Twenty-two participants with ALS and 28 healthy controls were screened. Initial scanning data were collected from 20 ALS and 20 control participants. Ten ALS and 14 control participants underwent repeat scanning at 6-month follow-up and 11 ALS and 13 control participants underwent repeat scanning at 12-month follow-up. A CONSORT flow diagram of the data collection is shown in Fig. 1.

The demographic and clinical features of participants are reported in Table 1. The ALS cohort was on average an early-stage cohort with mean ALSFRS-R of 40.0, with 60% in King's Stage 1 or 2. This proportion was reduced to 45.5% at the 1-year follow-up visit. As per revised El Escorial Criteria, there were seven possible, eight probable, and five definite subjects at baseline. Out of the seven possible subjects, one progressed to definite, two progressed to probable, three remained as possible, and one was withdrawn from the study at the 1-year follow-up. Out of the eight probable subjects, three remained as probable and five were withdrawn from the study. Only two out of the five definite subjects returned for the 1-year follow-up and they remained in the definite status. The mean change in ALSFRS-R of ALS participants at the 1-year follow-up visit was −4.8 points,

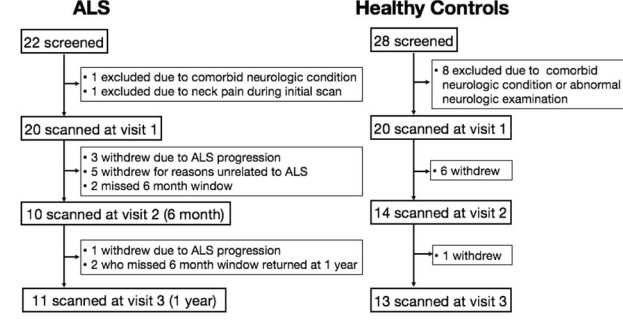

**Fig. 1 Flow diagram of study participation.** ALS participant withdrawals were recorded as being either due to ALS progression or due to reasons unrelated to ALS.

**Table 1 Demographic and clinical features of ALS and control cohorts at initial visit and at follow-up.**

| | Initial visit | | | Follow-up visit (1 year) | | |
| --- | --- | --- | --- | --- | --- | --- |
| | ALS | Control | *p*-Value | ALS | Control | *p*-Value |
| Sample size, *n* | 20 | 20 | | 11* | 13 | |
| Sex ratio, male : female | 11 : 9 | 10 : 10 | 1 | 7 : 4 | 6 : 7 | 0.44 |
| Age, years | 57.5 ± 9.8 (31–74) | 57.0 ± 8.7 (30–69) | 0.74 | 57.8 ± 11.2 (32–71) | 57.0 ± 9.6 (31–69) | 0.76 |
| Riluzole use, yes : no | 9 : 11 | — | | 6 : 5 | — | |
| Disease duration, months | 38.5 ± 42.5 (3.4–147.7) | — | | 50.1 ± 53.8 (15.8–160.0) | — | |
| Site of onset | Upper limb (10) Lower limb (5) Bulbar (5) | — | | Upper limb (4) Lower limb (5) Bulbar: (2) | — | |
| King's disease stage (number of participants) | Stage 1 (5) Stage 2 (7) Stage 3 (7) Stage 4 (1) | — | | Stage 1 (2) Stage 2 (2) Stage 3 (2) Stage 4 (5) | — | |
| El Escorial classification (number of participants) | Possible (7) Probable (8) Definite (5) | — | | Possible (3) Probable (5) Definite (3) | — | |
| ALSFRS-R total score (0 most severe–48 normal) | 40.0 ± 5.5 (27–45) | — | | 38.2 ± 4.9 (31–47) | — | |
| ALSFRS-R upper-limb sub-score (0 most severe–36 normal) | 29.7 ± 5.3 (15–36) | — | | 28.6 ± 4.3 (22–36) | — | |
| ECAS total score (0 most severe–136 normal) | 113.3 ± 7.9 (90–126) | 119.8 ± 8.7 (103–129) | 0.04 | 116.9 ± 7.2 (106–129) | 122.4 ± 7.2 (108–134) | 0.04 |
| ECAS ALS-specific sub-score (0 most severe - 100 normal) | 84.4 ± 7.6 (59–95) | — | | 86.2 ± 5.5 (79–97) | — | |
| UMN burden score (0 normal – 6 most severe) | 2.5 ± 1.3 (1–6) | — | | 2 ± 1.5 (0–4) | — | |

*Four deaths occurred in ALS participants before the 1-year visit; there were no deaths in controls.

with an average slope of −0.4 points/month. There were four deaths in the ALS cohort and no deaths in the control cohort during the study.

Visual analysis of segmentation, as described in "Methods," demonstrated up to 40% overlap in rubrospinal tract volume with the lateral CST. For this reason, rubrospinal tract data are not reported here.

**Cross-sectional data: FA along the cord.** Figure 2a shows the differences in mean FA between the ALS and control groups in the whole cord at the initial visit, with the group SDs shown as error bars (box plots of the same data are included as Supplementary Information, Supplementary Fig. 1). The greatest significance of differences is noted in the C2–C3 and C6 areas. Figure 2b shows the VCM-corrected *p*-value heat map, demonstrating statistically significant group differences (corrected *p* < 0.05) at C2–C3 and the inferior part of C6.

Figure 2c shows the mean FA plot and Fig. 2d shows VCM *p*-value heat map after excluding gray matter areas. Differences in the C2–C3 and C6 regions are more extensive in this case, compared with the whole cord analysis. Figure 2e shows the group differences in mean FA in the descending fiber tracts within the white matter and Fig. 2f shows the corresponding VCM *p*-value heat map. In this case, the FA differences are significant throughout the entire C2–C6 area (with corrected *p*-values < 0.001 at the C2 level). This demonstrates that segmentation of white matter substantially improves differentiation between ALS and control participants.

**Cross-sectional data: tract-specific FA along the cord.** We further segmented the data to extract individual spinal tracts, using the Spinal Cord Toolbox (SCT), and found statistically significant cross-sectional group differences in FA along almost the entire C2–C6 area in the lateral CST and to a less consistent degree in the spinal lemniscus (Fig. 3). We also noted a clear trend in the

group difference in FA in the posterior column, with a well separated group mean throughout C2–C6, but this difference was not statistically significant due to the large within-group variance.

**Cross-sectional data: other diffusion metrics.** The sensitivity of RD to differentiate participants with ALS from controls was less striking, but was also greatest with segmentation and analysis of descending tracts (Supplementary Fig. 2). The significance of the cross-sectional group differences in MD was marginal, and we did not find statistically significant differences in axial diffusivity (AD). We also found statistically significant cross-sectional differences at several levels in RD (Supplementary Fig. 3) and MD, but not in AD in lateral CST and spinal lemniscus. In all cases, the group differences were greatest in the rostral and caudal aspects of the cervical cord.

**Cross-sectional data: CSA along the cord.** Figure 4a shows group differences in CSA from C2 to C6. Significant differences are found (Fig. 4b) for the entire cervical cord length, which reflects cord atrophy from C2 to C6. We also analyzed the spinal white and gray matter separately, and identified significant group differences along the cord in CSA in both cases (Fig. 4c-f). Differences were greatest at the C2 level, with corrected *p* < 0.002, <0.009, and <0.008 for the whole cord, the white matter of the cord, and the gray matter of the cord, respectively.

Figure 5 provides a summary of FA and CSA findings for all participants. The dot plots show mean FA across C2–C6 and at C2 levels for whole cord, white matter, gray matter, descending tracts, lateral CST, and spinal lemniscus. Figure 5 also shows mean CSA of whole cord, white matter, and gray matter. The *p*-values of the differences from the VCM analysis are reported in Figs. 2–4. Significant differences between ALS and control participants were noted at the C2 level, especially for the descending tracts (*p* < 0.001) and the lateral CST (*p* < 0.007). An unexpected finding was the demonstration of significant

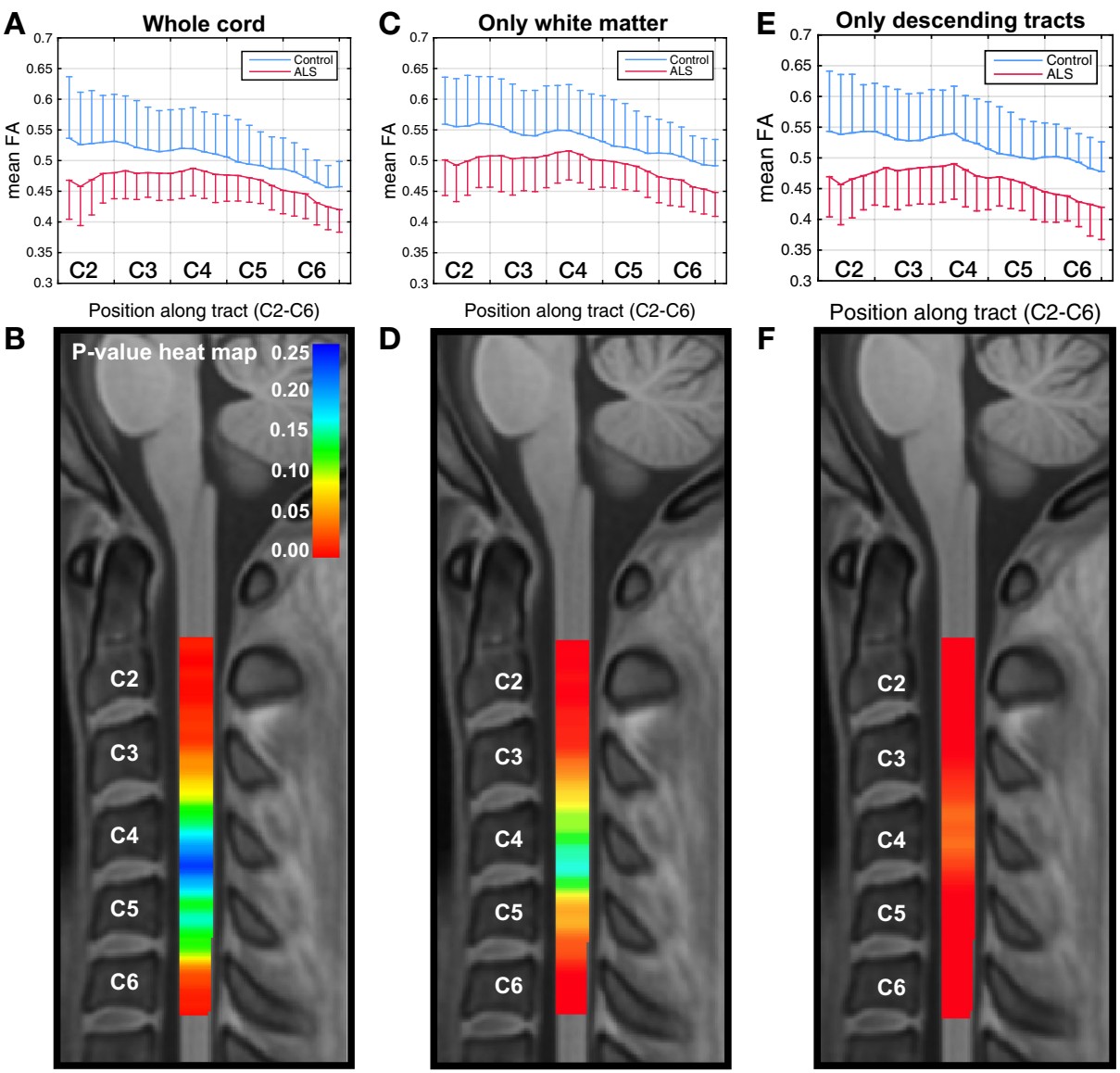

**Fig. 2 Cross-sectional differences in FA along the cord.** Graphical representations of cross-sectional differences in FA between ALS ($n = 20$) and control ($n = 20$) participants (the one-sided error bars represent SD), and corresponding VCM-corrected $p$-value heat maps, highlighting spinal levels with significant group differences for whole cord without segmentation, only white matter, and only descending tracts (corticospinal, rubrospinal, reticulospinal, lateral vestibulospinal, and tectospinal). In the $p$-value heat map, colors red to orange represent $p$-values $< 0.05$ and colors yellow to blue represent $p$-values $> 0.05$. The sensitivity of the method increases with segmentation. Box plots of the same data are included as Supplementary Information, Supplementary Fig. 1.

differences at C2 in the spinal lemniscus ($p < 0.02$). When all the slices from C2 to C6 are considered, the descending tracts ($p < 0.04$) maintained significant differences between ALS and controls.

**Cross-sectional data: multimodal analysis of dMRI and CSA.** As FA and CSA both differentiate ALS from control participants in our cohort, but may represent complementary information, we performed a multimodal analysis combining FA and CSA. We found that combining whole cord FA and CSA provides an extremely sensitive metric to the disease effect captured in the cross-sectional data, with a significant difference between groups (VCM-corrected $p < 0.0065$) compared with $p < 0.23$ and $0.02$ with the separate analysis of FA and CSA, respectively. The VCM-corrected $p$-values at the C2 level were found to be $<0.0003$ with the multimodal analysis of FA and CSA compared

with $p < 0.02$ and $0.002$ with the separate analyses of FA and CSA, respectively.

**Cross-sectional data: analysis based on site of onset.** Although our sample size provides limited power to conduct subgroups analyses, we performed an exploratory analysis to study whether the site of onset had any influence on the spinal cord FA and CSA metrics. We analyzed FA along the lateral CST and CSA along the whole cord for the subgroups with upper-limb onset (10 participants), lower-limb onset (5 participants), and bulbar-onset ALS participants (5 participants), and compared it with that of the 20 control participants (Supplementary Fig. 4). We noted that the upper-limb onset participants had lower mean FA (0.45) and CSA ($56.7 \, mm^2$) compared with lower-limb onset participants ($0.47$ and $58.4 \, mm^2$ respectively) and the lower-limb onset participants had lower mean FA and CSA compared with

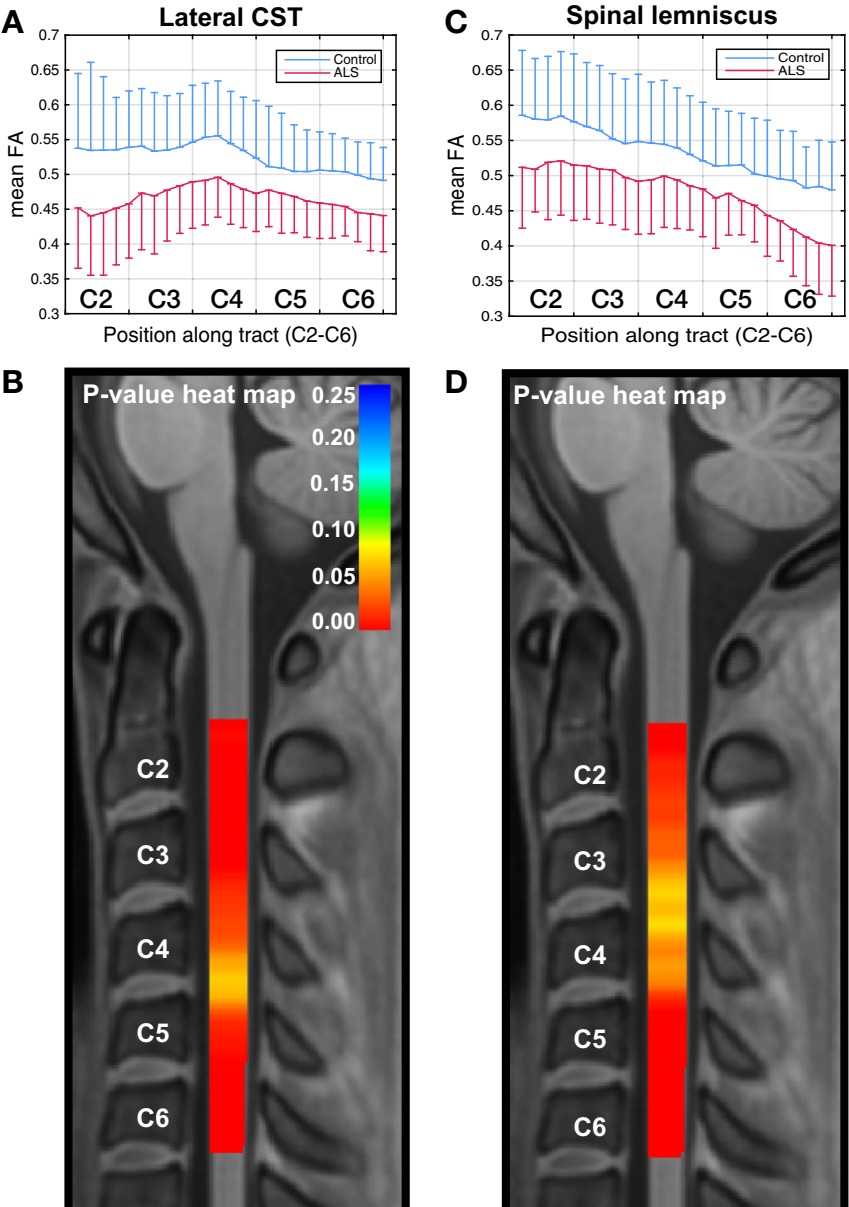

**Fig. 3 Cross-sectional differences in tract-specific FA along the cord.** Graphical representations of cross-sectional differences in FA between ALS ($n =$ 20) and control ($n = 20$) participants (the one-sided error bars represent SD), and corresponding VCM-corrected $p$-value heat maps, highlighting spinal levels with significant group differences for lateral CST and spinal lemniscus. In the $p$-value heat map, colors red to orange represent $p$-values < 0.05 and colors yellow to blue represent $p$-values > 0.05. Box plots of the same data are included as Supplementary Information, Supplementary Fig. 1.

bulbar-onset participants (0.50 and 65.5 mm² respectively). However, these differences did not reach statistically significant level. The control participants had the highest mean FA (0.53) and CSA (69.0 mm²).

**Cross-sectional data: correlations with functional status.** We found a strong correlation between mean whole cord FA values and the ALSFRS-R spinal sub-score (questions 4–12, Fig. 6a), with the highest correlation attained at the C2 level (Fig. 6b). We also found a strong correlation between the mean whole cord CSA and the ALSFRS-R spinal sub-score at the C2 level (Fig. 6d). We found that white matter CSA at C2 correlates more strongly with the ALSFRS-R sub-score than gray matter CSA (Fig. 6e, f, respectively).

The whole cord C2–C6 mean FA and the upper motor neuron (UMN) burden score demonstrated a significant negative correlation (correlation coefficient = −0.51, $p$-value = 0.02). We did not find significant correlation of whole cord, white matter, or gray matter CSA with the UMN burden score.

We also noted post-hoc that gray matter FA at C2 at initial visit predicted study withdrawal due to progression of functional deficits from ALS. We found significant differences ($p$-value 0.017) in gray matter FA at C2 between 5 ALS participants who withdrew from the study due to disease progression (mean 0.47, range [0.42–0.51], SD: 0.03) and the other 15 ALS participants (mean 0.56, range [0.45–0.62], SD: 0.05). This finding is also consistent with our finding that the correlation between FA and ALSFRS-R is strongest at C2.

Finally, we analyzed the possible correlation of FA and CSA at C2–C3 level with respiratory function (measured with questions 10–12 of ALSFRS-R), we did not find any significant correlations between these measures and the respiratory function.

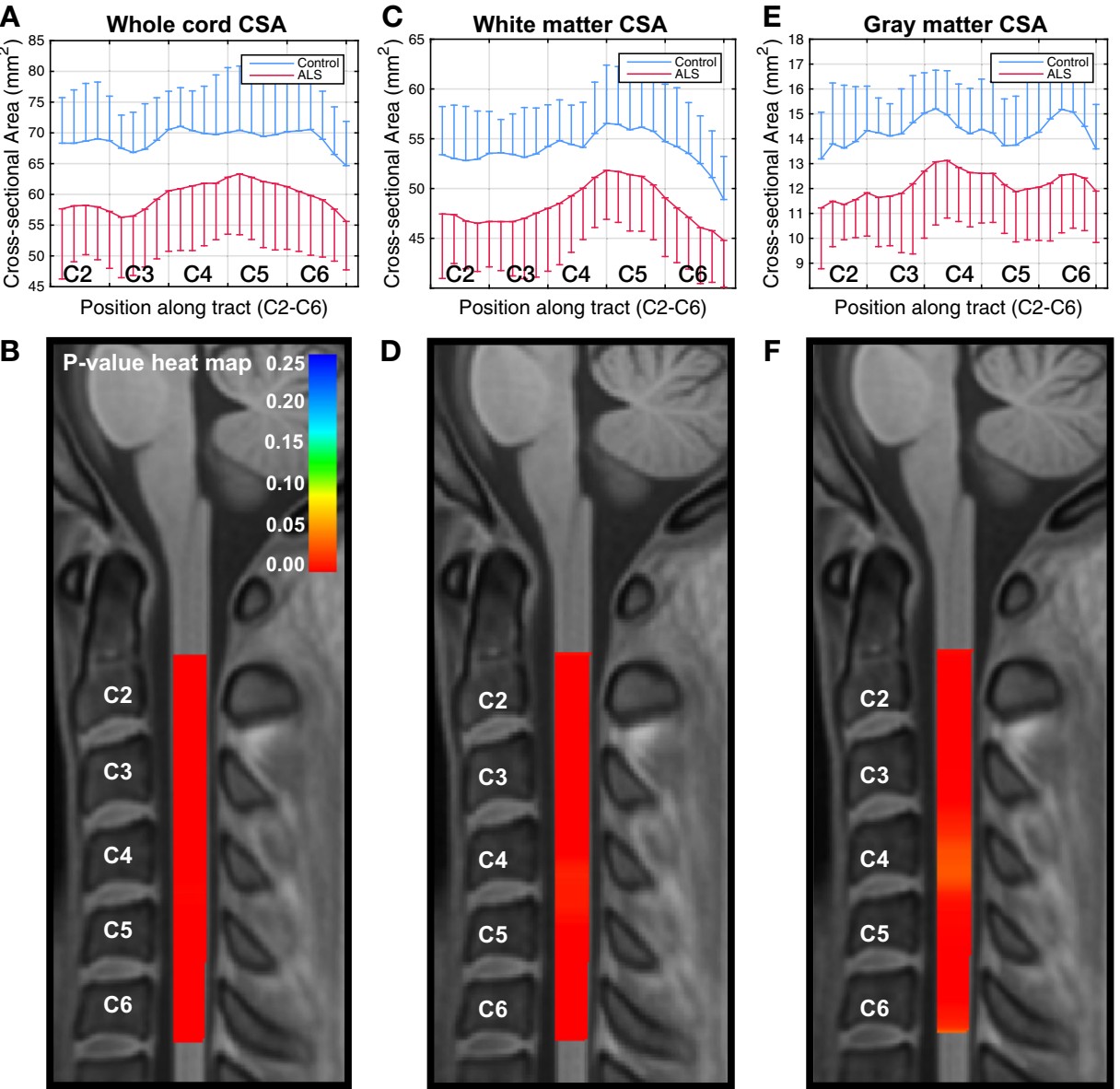

**Fig. 4 Cross-sectional differences in CSA along the cord.** Graphical representation of cross-sectional differences in CSA between ALS ($n = 20$) and control ($n = 20$) participants (the one-sided error bars represent SD), and corresponding VCM-corrected $p$-value heat maps, highlighting spinal levels with significant group differences for whole cord, white matter, and gray matter. In the $p$-value heat map, colors red to orange represent $p$-values < 0.05 and colors yellow to blue represent $p$-values > 0.05. Box plots of the same data are included as Supplementary Information, Supplementary Fig. 1.

**Longitudinal data: tract-specific dMRI averaged across C2–C6.** Figure 7 shows the mean and SD of the initial and 1-year follow-up C2–C6 mean diffusion metrics of 11 ALS participants. We found a statistically significant longitudinal increase in RD and MD in all analyses (whole cord, only white matter, only descending tracts, lateral CST, spinal lemniscus, and the posterior column). We did not find significant longitudinal changes in FA. There were no statistically significant longitudinal changes in diffusion metrics, in any of the tracts, in the 13 control participants scanned at one year (Supplementary Fig. 5).

We did not find significant changes in diffusion metrics from the initial visit to the six-month follow-up visit for either ALS or control participants. For the ALS participants, we noted a trend for decreased FA in the lateral CST ($p = 0.06$) and the posterior column ($p = 0.08$) at six months.

**Longitudinal data: tract-specific dMRI along the cord.** Segmental analysis of the group difference in 1-year longitudinal change (i.e., change in ALS vs. change in control) showed significant group differences in change in RD and change in MD from C4 to C6 (both increased in ALS participants). Significant differences were noted for the combined descending tracts and the lateral CST (Fig. 8), but not for the other tracts studied. The increase in these metrics over time was ~10% in the ALS participants in the levels where significant group differences in longitudinal change were noted.

Finally, we analyzed the baseline cross-sectional difference in FA along the lateral CST for only the 11 ALS participants we followed up at 1 year (Supplementary Fig. 6). Despite representing a smaller sample size than the full ALS cohort, and hence one with less statistical power, this subgroup analysis still

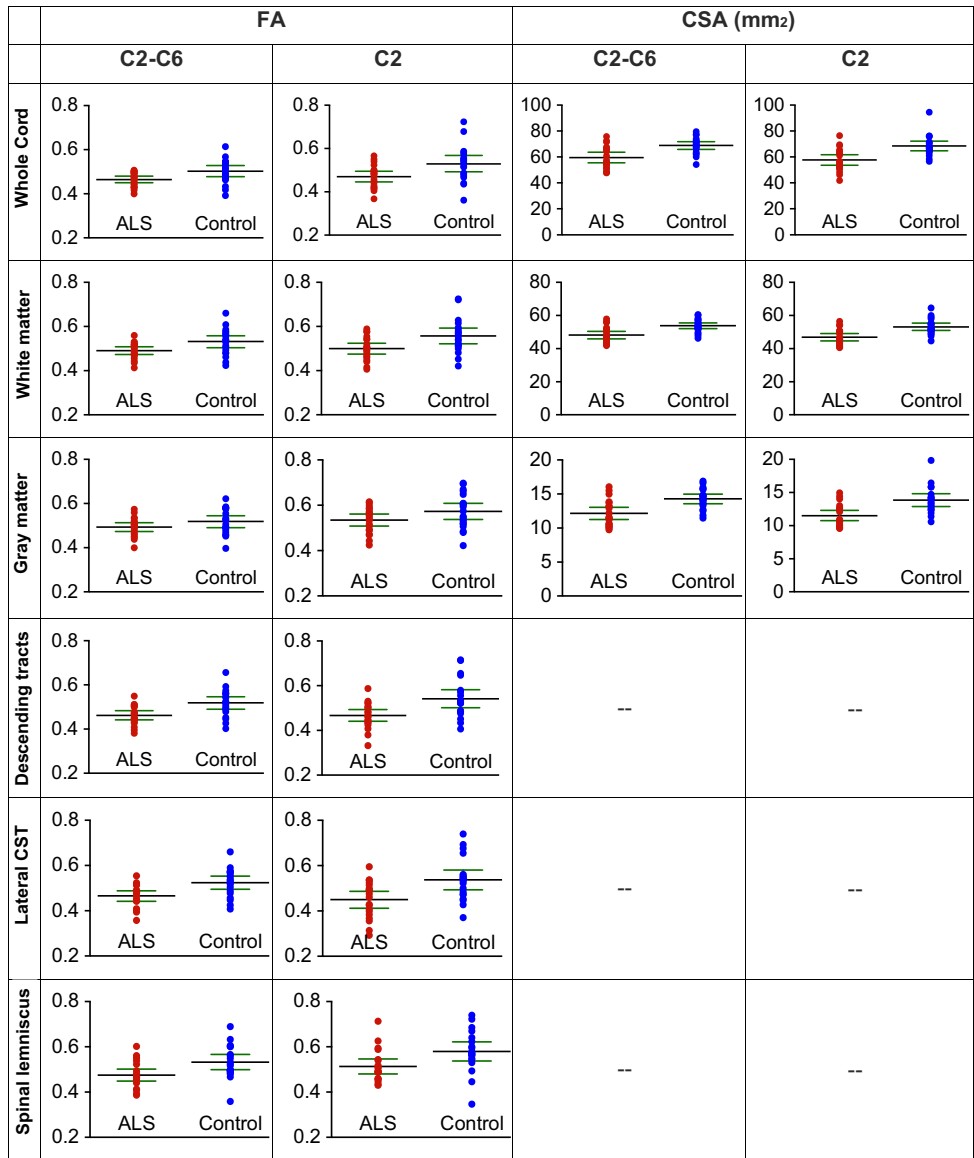

**Fig. 5 Summary of FA and CSA data of whole cord, white/gray matter, and individual tracts.** The dot plots show the mean values (black lines) with 95% confidence intervals (green lines). Sample size $n = 20$ for ALS and control participants.

demonstrated a statistically significant reduction in FA from C2 to C3 compared with control subjects.

**Longitudinal data: cord CSA**. We did not identify statistically significant change in spinal cord CSA (whole cord, white matter, or gray matter), along or averaged across the cord, at 6 or 12 months.

**Longitudinal data: correlations with functional status**. We did not find a significant correlation between the change in any of the diffusion metrics or CSA (whole cord, white matter, or gray matter), and the change in ALSFRS-R spinal sub-score over a 1-year period.

**Discussion**
In this study, we present findings of tractography along the cord and extracted diffusion metrics along spinal pathways to analyze their continuous variation from C2 to C6 in people with ALS and

an equivalent number of healthy control participants at enrollment and longitudinally over 1 year. This tractography-based approach facilitates between-subject statistical analysis tailored to the specific anatomy of each participant[14], potentially resulting in more accurate inter-group comparison. The streamline tractography-based method, combined with VCM, allowed us to study the continuous association between disease status and diffusion properties along the cord, contrary to prior parametric approaches, which focused only on the mean whole cord data from each discrete spinal level. Our data also demonstrate that analysis of individual fiber tracts increases the sensitivity of diffusion metrics to disease-related changes, strengthening the potential of spinal cord diffusion magnetic resonance imaging (dMRI) as a biomarker of disease progression. We found the greatest differentiation from controls in the lateral CST and spinal lemniscus.

Among diffusion metrics, FA was the most sensitive as a cross-sectional biomarker, whereas RD and MD were sensitive to longitudinal change. The lack of significant longitudinal change

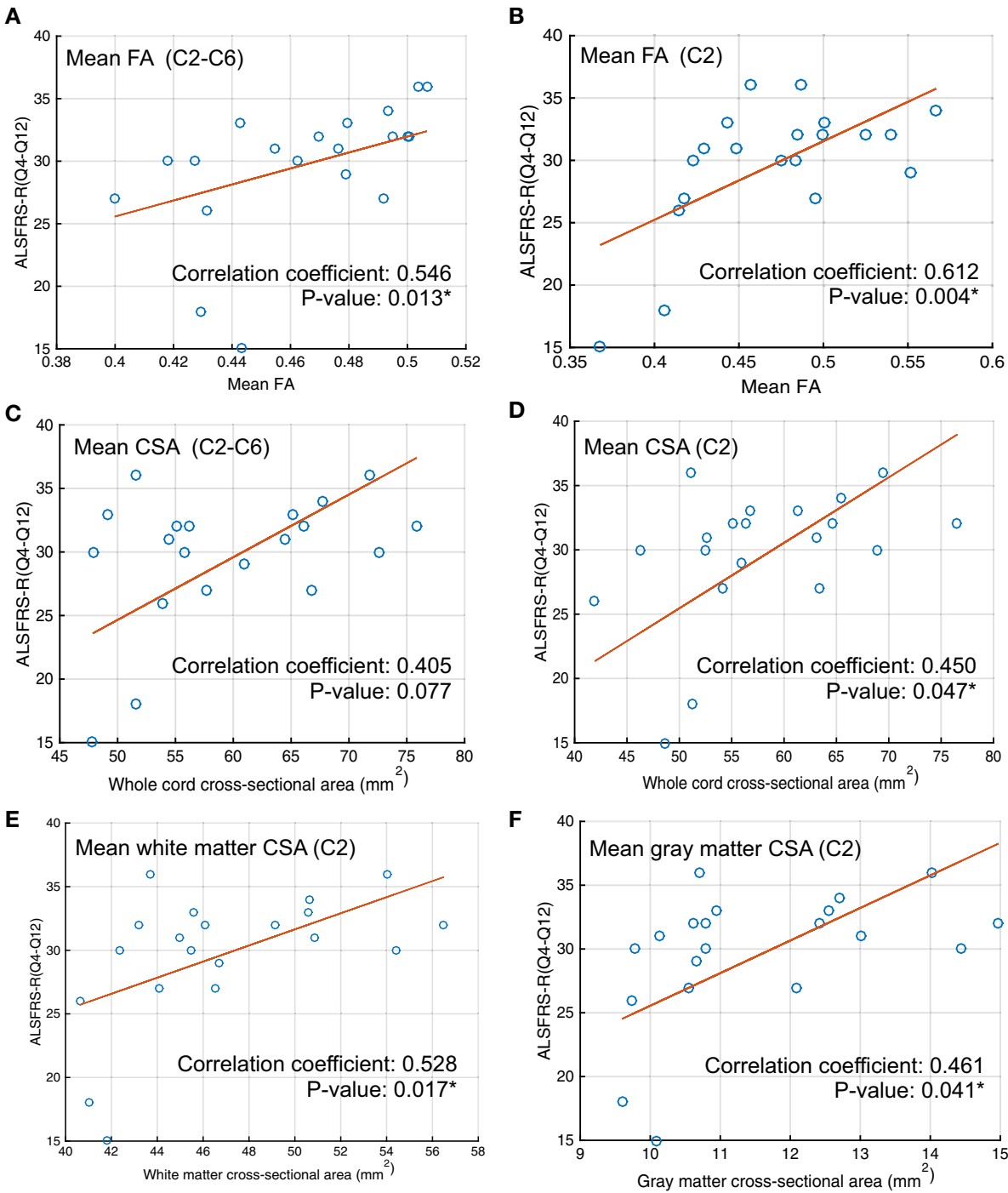

**Fig. 6 Correlation analysis.** Correlation between ALSFRS-R (Q4-Q12) and (**a**) mean whole cord FA from C2 to C6, (**b**) mean whole cord FA at C2, (**c**) mean whole cord CSA from C2 to C6, (**d**) mean whole cord CSA at C2, (**e**) mean white matter CSA at C2, and (**f**) mean gray matter CSA at C2. Sample size $n = 20$.

in FA agrees with the report by Mendili et al.[10] (cross-sectional results are not reported in their work). Compared with FA, MD in CST has shown a stronger trend in longitudinal change as per their report, similar to the higher sensitivity of MD and RD we observed. However, no significant longitudinal changes in any of the diffusion metrics were reported in their work.

Our ALS cohort has a relatively slow progression of clinical functional loss. With regard to other clinical characteristics, such as age (mean 57.5 years), site of onset (majority limb onset), and gender (slight majority male), our cohort is typical of the ALS

population. The sole atypical feature, slow progression, in fact highlights the value of our findings. Despite a small sample size of largely slow progressors, we observed a statistically significant longitudinal change in RD and MD in numerous analyses of ALS subjects, with our tract-specific approach. This suggests that our method will be highly sensitive to change in a more typical cohort and possibly over a shorter period of observation.

Segmentation analyses in our cross-sectional study demonstrate that, as expected from the principal clinical and pathologic features of ALS, focusing on descending tracts enhances our

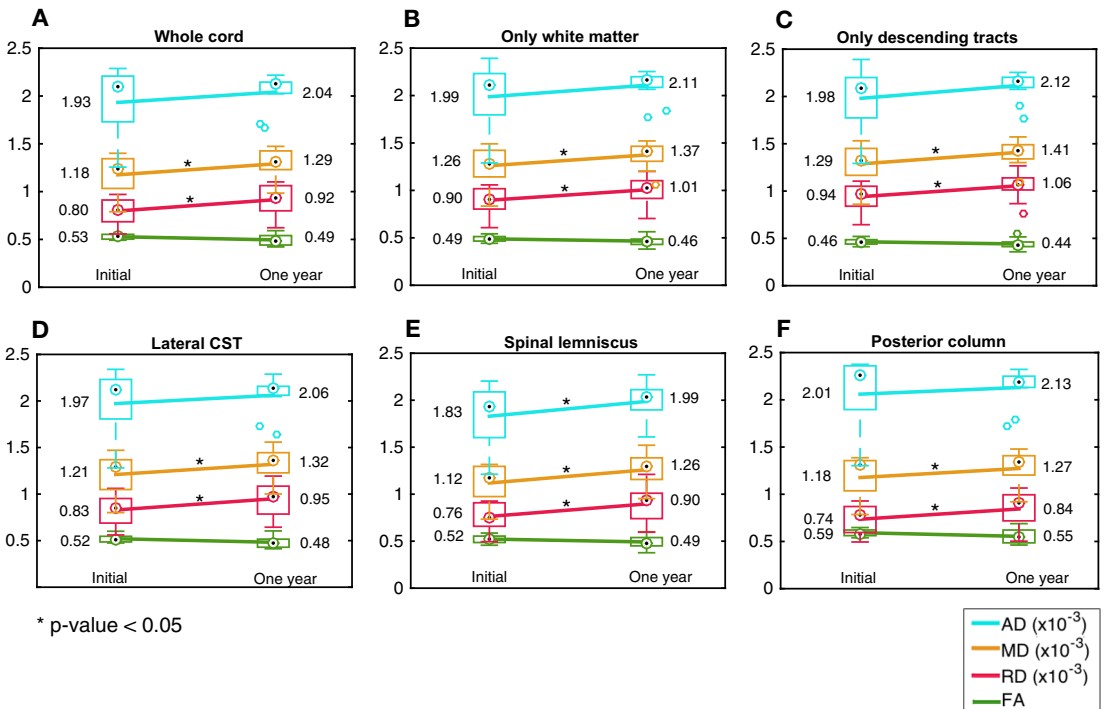

**Fig. 7 Longitudinal changes in diffusion metrics.** Longitudinal changes in ALS participants ($n = 11$), in FA, RD, MD, and AD (averaged across C2–C6) in (**a**) whole spinal cord, (**b**) spinal white matter, (**c**) descending tracts, (**d**) lateral CST, (**e**) spinal lemniscus, and (**f**) posterior columns. The line between initial and one year data connects the means (the numerical values in the text). The box plots show the median (circle with dot), 25th and 75th percentiles (edges of the box), and the range of FA/CSA (the whisker end points) excluding the outliers (individual circles).

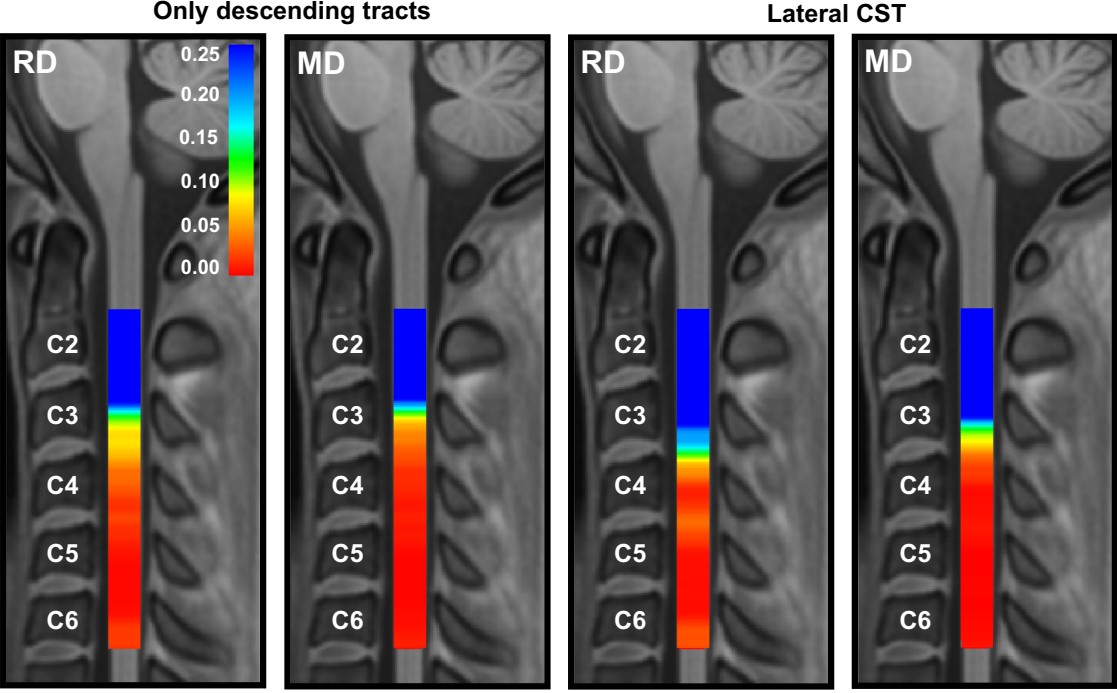

**Fig. 8 Graphical representation of the VCM-corrected *p*-value heat maps of group difference in longitudinal changes in RD and MD, in the 1-year follow-up data.** Significant increase, with a size of ~10%, is noted in the group of descending tracts and the lateral CST. In the *p*-value heat map, colors red to orange represent *p*-values < 0.05 and colors yellow to blue represent *p*-values > 0.05.

ability to differentiate ALS from controls. In addition, apparent involvement of the spinal lemniscus and longitudinal changes seen in this tract and the posterior columns suggest some involvement of sensory pathways, particularly as ALS progresses.

This is consistent with the observation that the post-central gyrus is affected as TDP-43 pathology spreads in ALS[15]. Our observations of reduced FA in ascending tracts is also consistent with findings by Iglesias et al.[16] and Cohen-Adad et al.[17] of

impairment of sensory pathways in ALS, and with prior work indicating abnormal somatosensory-evoked potentials in ALS[18]. The involvement of sensory pathways in ALS was also suggested in autopsy[19] and electrodiagnostic studies[20].

We observed a consistent spatial decrease in FA (negative slope across the spine, Fig. 2a) and increase in RD (positive slope in Supplementary Fig. 2a) from C2 to C6 in healthy controls. Rasoanandrianina et al.[11] noted a similar trend, with higher FA and lower RD at the C2 level compared with the C5 level. This may reflect gray matter expansion and/or crossing fiber configurations due to fascicles leaving and entering the cord in the cervical enlargement.

In addition, we show a striking difference in cord CSA between individuals with ALS and healthy control participants. The combined analysis of FA and CSA shows the potential for multimodal spinal cord investigations to increase the sensitivity of imaging data.

In our longitudinal study, we found consistent changes in RD and MD, and a parallel trend in AD that did not reach statistical significance, in the whole cord aand in individual tracts, in ALS subjects. This is particularly notable in light of the slow progression of functional loss and the limited sample size of our cohort.

Given the strong differences in FA in the cross-sectional data, it may be surprising that RD and MD appear more sensitive to longitudinal change over 1 year's observation. This may be due to the reduced variance and hence higher statistical power, of RD and MD compared with FA. Alternatively, RD and MD may be more sensitive to specific pathophysiological changes at a later stage of the disease. Increased RD and MD may represent the enlargement of extracellular spaces due to axon loss, which can promote water diffusion, as pointed out by Agosta et al.[9], based on their observation of increased MD in a cohort of 17 subjects. Notably, the rate of functional decline in our cohort was slower than has been reported previously in ALS[21], with several participants demonstrating little or no progression over the course of the study. This may also contribute to the fact that only a trend toward reduction in FA was noted in our longitudinal data.

In addition to its potential utility as a biomarker in ALS, tract-specific imaging provides the potential for advancing our understanding of the pathology of ALS longitudinally and in vivo. The demonstration of cross-sectional differences in FA beyond the lateral CST and the longitudinal changes in RD and MD in afferent pathways that were not seen in healthy control participants, raise important questions. As noted above, previous pathological, neurophysiological, and imaging data demonstrate convergent evidence of degeneration beyond the primary motor pathways in ALS. Specifically, there is established evidence to suggest involvement of large myelinated fibers, the posterior columns, and primary sensory cortex[20]. Our findings add to emerging evidence of involvement of sensory pathways, both posterior columns and spinal lemniscus, in ALS, providing further support to the hypothesis that ALS pathology spreads from primary motor systems to become a widespread disorder of the central and peripheral nervous systems.

The higher mean FA and CSA in the C2–C6 region of the bulbar-onset ALS participants, compared with the limb-onset participants, may reflect less involvement of the spinal cord in bulbar-onset disease, although mean FA and CSA of bulbar-onset participants are still lower than those of controls. Similarly, the lower mean FA and CSA in the C2–C3 region of the upper-limb onset participants, compared with lower-limb onset participants, may reflect more severe involvement in both the long tracts and the segmental fascicles in these participants. However, we note that these analyses were done with a limited number of participants in each subgroup (10 upper limb, 5 lower limb, and 5 bulbar). Further studies with more participants are therefore required to confirm these preliminary results.

Although we acquired high-quality data corrected for multiple artifacts, with relatively high in-plane axial resolution of $1.12 \times 1.12 \, mm^2$, the process of segregation of fiber tracts remains challenging. This is primarily because the fiber tracts are in fact not as discrete as they appear in anatomic atlases and axons from functionally different tracts may actually overlap. This overlap between the tracts makes it difficult to assess that the changes observed in some pathways are, in fact, unique to that particular pathway. To address this issue, we visually checked all the segmentations and excluded those voxels in segmentations which had a confidence value of <0.1 (our findings are not sensitive to this threshold, as long as the same threshold within a recommended range of [0.0–0.2] is used for all the subjects). This overlap is not a meaningful issue in the posterior columns, as there is no contamination by adjacent fiber pathways, and is minimal in the lateral CST and spinal lemniscus due to the relatively higher volume of these tracts. However, we excluded the data from rubrospinal tract, due to its possible contamination by the data from lateral CST.

Our study, like others, is limited by relatively small participant numbers and high dropout over the course of 1 year. This remains a shortcoming of ALS-imaging studies with relatively long periods of follow-up.

Our cross-sectional findings, as noted by others, may have value as a prognostic biomarker. For imaging to be useful as a biomarker of disease progression, robust changes that pre-date changes in functional status will need to be demonstrated. Our longitudinal data demonstrate only a trend at 6 months and a statistically significant change over a 12-month period. However, it is worth noting that this was identified in a small longitudinal cohort of 11 participants with unusually slowly progressive disease, as judged by an ALSFRS-R slope of 0.40 points/month compared with the reported average change of 0.9 to 1 points/month[21]. This suggests that diffusion metrics may have the potential to provide robust biomarkers of progression in a cohort of the size typical in a phase 2 clinical trial.

Future work in this area should include a longitudinal assessment in a larger and phenotypically broader ALS cohort, including people with a more typical rate of disease progression. The imaging metrics described here may also prove useful in providing supportive evidence of CST degeneration in lower motor neuron-predominant ALS to differentiate it from motor neuropathies that can mimic ALS. We also anticipate future studies correlating diffusion metrics in afferent pathways with clinical and neurophysiological markers of sensory dysfunction, so as to better and more fully characterize the pathophysiology of ALS. Finally, our observation that FA at C2 predicts study withdrawal due to disease progression indicates that diffusion metrics at this level merit further evaluation as possible prognostic biomarkers in ALS.

## Methods

**Study participants and design.** People who met revised El Escorial Criteria[22] for clinically possible, probable, or definite ALS were recruited from the ALS Association Certified Treatment Centers of Excellence at the University of Minnesota and Hennepin County Medical Center. Healthy control volunteers were recruited from the general public and selected to match ALS participants' age range and sex frequency. Exclusion criteria included the presence of neurologic illnesses other than ALS, the inability to tolerate MRI scanning, and the failure to meet MRI safety criteria. Participants were enrolled after providing written informed consent using procedures approved by the Institutional Review Board: Human Subjects Committee of the University of Minnesota. All participants underwent imaging at time of enrollment, in compliance with all the ethical regulations, and were asked to return for follow-up visits at 6 and 12 months after enrollment. We recently reported magnetic resonance spectroscopy brain findings in this cohort[23,24].

**Image processing and feature extraction pipeline**

**Fig. 9 MRI data processing and analysis. a** Flow diagram depicting the image processing and feature extraction pipeline. **b** Representative results of streamlines obtained by tractography in the whole cord and segmentations of the cord into white/gray matter and individual tracts.

**Clinical assessments**. Each participant underwent a neuromuscular examination by a neuromuscular neurologist (D.W., G.M., and G.G.) at enrollment and at the 12-month follow-up. Functional impairment in ALS participants was measured at enrollment, 6-month, and 12-month visits using the ALSFRS-R[25]. Clinical staging was done using an algorithm developed by King's College London (stage 1 mild–stage 4 advanced)[26]. Cognitive and behavioral status was assessed in all participants at all visits using the Edinburgh Cognitive Behavioral ALS Screen (ECAS)[27]. Scores on the total ECAS and its ALS-specific component were recorded. A UMN burden score was derived based on the neuromuscular examination as previously described[23] and ranged from 0 to 6, with a higher score indicating greater UMN burden. Current riluzole use was also documented. Disease duration was calculated as the time since the date of first reported symptoms to the date of the MRI exam. All clinical assessments were performed within 1 week of the MRI exam.

**MRI data collection**. dMRI data were acquired using 3T Siemens Trio and Prisma scanners at the Center for Magnetic Resonance Research at the University of Minnesota. Vertebral levels C2–C7 were scanned (we later excluded the C7 level from the analysis, due to partial coverage in some subjects). The data were obtained using the RESOLVE sequence[28], with diffusion gradients along 30 directions and a b-value of 650 s/mm$^2$. Six additional volumes without diffusion encoding were equally interleaved in the dataset yielding a total of 36 volumes. We obtained 30 slices with thickness 3.3 mm and voxel size $1.12 \times 1.12$ mm$^2$ (FoV = $118 \times 62$). Two sets of data were collected during each session, with reversed phase encoding directions (anterior to posterior and posterior to anterior). They were subsequently combined to correct for distortions and to increase signal to noise ratio[29].

**MRI data processing**. The data were corrected for distortions due to eddy currents, susceptibility-induced off-resonance artifacts, and subject motion[29,30]. We excluded the C7 level from the analysis, due to partial coverage in some subjects and increased distortions in this region located at the edge of the field of view. We also manually checked and adjusted all datasets to ensure proper correspondence and alignment of the C2–C6 levels across all subjects. A DTI model was subsequently fitted to the corrected data using FSL[30], and streamlines (i.e., approximate white matter fiber tracts represented by three-dimensional line segments) were extracted using a deterministic tractography algorithm available in the diffusion toolkit[31]. The DTI metrics FA, RD, MD, and AD were extracted from each location along each streamline. A sliding mean window with a width of five points (~6 mm), two points superior and two points inferior to the current point, was used to estimate the feature at each location on each streamline, and to mitigate the effects of noise. Segmentation of the spinal cord into white matter and gray matter, and into individual pathways (e.g., lateral CST) within the white matter[32], was done using a probabilistic method from the SCT[33]. The mean of each diffusion metric, from all tractography streamlines passing through each segmented pathway, was computed at each location along the cord, yielding a spatially continuous one-dimensional characterization of the microstructure of each spinal pathway

(e.g., one FA vector from C2 to C6 for the lateral CST). Similarly, the CSA of the whole cord was computed at each location.

Figure 9a summarizes the data processing steps. Figure 9b shows representative results of the extracted whole-cord streamlines, and cord segmentations. We first segmented the cord into white and gray matter, and then into ascending and descending pathways within the white matter. The descending pathways included in the analysis are the corticospinal (lateral and ventral), rubrospinal, reticulospinal, lateral vestibulospinal, and tectospinal tracts. The ascending pathways included the posterior column, ventral spinocerebellar tract, and the spinal lemniscus. We analyzed each of the above-mentioned individual tracts independently, averaging data from left and right sides for all the tracts, and performed collective analyses of descending and ascending tracts. To address possible overlap between fiber tracts we visually checked all the segmentations and excluded those voxels in segmentations, which had a confidence value of <0.1.

**Statistical analysis of single-modality cross-sectional data**. We used a VCM[13], which was introduced for the functional analysis of diffusion tensor tract statistics, for the analysis of our cross-sectional data. The multivariate VCM is given by

$$Y_{i,j}(s) = x_i^T B_j(s) + \eta_{i,j}(s) + \varepsilon_{i,j}(s) \tag{1}$$

where $Y_{i,j}(s)$ is the jth imaging metric (e.g., FA, CSA) of the ith subject at position s along the cord; $x_i$ is a vector of person-level characteristics and contained disease status, age, and scanner type (Trio, Prisma); $B_j(s)$ is the varying coefficient vector (one coefficient per covariate); $\eta_{i,j}(s)$ characterizes the individual curve variations (i.e., subject- and location-specific variations); and $\epsilon_{i,j}(s)$ is the measurement error. $^T$ denotes a vector transpose.

The VCM characterizes the associations between diffusion metrics (across tract positions, s) of each pathway, and covariates of interest: disease status, age, and the scanner used. The varying coefficients were estimated using a weighted least-squares method. A local test statistic (refer Eq. (9) in ref. [13]) was defined to assess whether the local varying coefficient for disease status, at each position along the cord, deviates significantly from zero. The null hypothesis was that local varying coefficients are zero (i.e., there exists no inter-group difference in the imaging metrics of the ALS and control participants at each position along the cord). The local significance (p-value) at each spatial location, between C2 and C6, is reported. The p-values were corrected for multiple testing along the cord[13]. The p-value calculation is done with a wild bootstrap method, by checking whether the local test statistic at each tract location is greater than the maximum of the test statistics at all grid points along the tract (i.e., the number of multiple tests), for all the bootstrap samples, which corrects for multiple testing along the cord[13].

**Statistical analysis of multimodal cross-sectional data**. We also conducted a multimodal analysis for the combined effect of FA and CSA. The model used for this multimodal analysis is same as in Eq. (1), with two imaging metrics as

variables, given by

$$\begin{pmatrix} Y_{i,\mathrm{FA}}(S) \\ Y_{i,\mathrm{CSA}}(S) \end{pmatrix} = \begin{pmatrix} \chi_i^T B_{\mathrm{FA}}(S) \\ \chi_i^T B_{\mathrm{CSA}}(S) \end{pmatrix} + \begin{pmatrix} \eta_{i,\mathrm{FA}}(S) \\ \eta_{i,\mathrm{CSA}}(S) \end{pmatrix} + \begin{pmatrix} \varepsilon_{i,\mathrm{FA}}(S) \\ \varepsilon_{i,\mathrm{CSA}}(S) \end{pmatrix}. \quad (2)$$

In this case, the null hypothesis was that both FA and CSA varying coefficients for disease status (components of the vectors $B_{\mathrm{FA}}$ and $B_{\mathrm{CSA}}$ corresponding to the disease covariate) are zero, representing the absence of inter-group differences in either of these metrics.

**Analysis based on site of onset**. To study whether the site of onset has any influence on the spinal cord, we divided the ALS cohort into three subgroups based on the site of onset: upper-limb onset (ten participants), lower limb onset (five participants), and bulbar onset (five participants). We then analyzed the FA and CSA of each subgroup. FA along the lateral CST and CSA of the whole cord is reported. We also report the mean FA and CSA of these subgroups, and compare them with those of the control group (20 participants).

**Statistical analysis of single-modality longitudinal data**. To investigate whether there were longitudinal changes from initial to one year follow-up scans, we first conducted an analysis of the within-person means across all slices, C2 to C6, and streamlines for each pathway. Using each person's mean at baseline and mean at follow-up, we used two-sample two-tailed paired $t$-tests for each of the ALS and control groups. We also tested the longitudinal change in ALS participants against the longitudinal change in control participants from the within slice across-streamline means along the spinal cord, using the VCM. This is done by replacing the imaging metric in Eq. (1) with the change in imaging metric, defined as the difference in the metric between initial and 1-year follow-up data.

**Correlations with functional status**. We studied correlations of mean FA and of mean CSA, averaged over C2–C6 slices and averaged over C2 slices, with ALSFRS-R of the 20 ALS participants at initial visit. As corticobulbar projections are not in the spinal cord, we recomputed ALSFRS-R to exclude questions relating to bulbar function (i.e., including only questions 4–12). We also studied correlations between the within-person change in these metrics and the within-person change in ALSFRS-R sub-score (questions 4–12) over a 1-year period. To study a possible correlation between FA and CSA at C2–C3 level with the respiratory function (as one of the muscle innervated by this area is the diaphragm), we also analyzed the correlation of these metrics with respiratory function measured with questions 10–12 of ALSFRS-R.

**Statistics and reproducibility**. The statistical analysis is conducted as described in the previous subsections. Sample size and other statistical parameters are detailed in Table 1. The information to reproduce the experiments and results is provided in the Reporting Summary.

**Reporting summary**. Further information on research design is available in the Nature Research Reporting Summary linked to this article.

## Data availability
Deidentified data will be made available on request for the purposes of reproducing the results presented, subject to institutional approval. Source data used to generate the charts presented in the manuscript is found in Supplementary Data 1.

## Code availability
The code used to produce the results reported in this article is available at https://www.cmrr.umn.edu/downloads/alstract/.

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

## Acknowledgements

This study was partly supported by funding from the Bob Allison Ataxia Research Center, the Institute for Translational Neuroscience, the Curt O'Hagan ALS/PLS and ALS-Lou Gehrig funds of the University of Minnesota Foundation, NIH grants P41 EB015894, P41 EB027061, P30 NS076408, and NSF Physics of Living Systems grant PHY 1305537. We are grateful to the persons living with ALS, their families, and control participants who volunteered their time; without them this work would not be possible. We acknowledge Valerie Ferment, the clinical study coordinator, and Susan Rolandelli, RN, for study coordination, and Pamela Droberg, NP, for her assistance in recruitment. We also thankfully acknowledge the help received from Dr. Julien Cohen-Adad (Polytechnique Montréal) for customizing the spinal cord toolbox to segment the spinal cord from our diffusion data.

## Author contributions

P.K.P., L.E.E., D.W., and C.L. contributed to the study concept and design. P.K.P., L.E.E., I.C., G.M., G.G., H.B.C., M.B., D.W., and C.L. contributed to the data acquisition and analysis. P.K.P., L.E.E., M.B., D.W., and C.L. contributed to drafting a significant portion of the initial manuscript and figures, and all authors contributed to revision of the paper and to the final draft. D.W. and C.L. are joint senior authors.

## Competing interests

The authors declare no competing interests.
