## [Peer Review File · Communications Biology]

Reviewers' comments:

Reviewer #1 (Remarks to the Author):

This manuscript reports the results of a cervical cord diffusion MRI study performed on 20 ALS patients, 11 of whom also underwent follow-up for up to one year from baseline. Both baseline cross-sectional and longitudinal analyses of whole-cord and tract-specific diffusion MRI metrics, as well as cervical cord cross-sectional area were assessed. Patients were compared with 20 age- and sex-matched healthy controls (13 of which underwent follow-up at 1 year). A semi-automated procedure for the segmentation of cord tracts was used, demonstrating increased sensitivity of mean FA in distinguishing ALS patients from healthy subjects in the cross-sectional analysis, particularly when only the descending tracts were considered. The 1-year longitudinal analysis showed significant progression of RD and MD in ALS patients, whereas these values remained stable in healthy controls. Significant correlations between FA and measures of clinical impairment were also found in the cross-sectional analysis.

The main novelty of this study is the application of a semi-automated method for the definition of specific tracts within the cervical cord using diffusion MRI data, in order to identify sensitive biomarkers of ALS progression that do not depend on the delineation of ROIs by an operator. Moreover, only a few longitudinal studies assessed cervical cord alterations in ALS. However, the results are not ground-breaking per se (as they are in line with previous reports) and the sample size is small, therefore reducing the significance and complicating the interpretation of the results, particularly when cross-sectional and longitudinal results are compared. The clinical features of the included sample are also atypical, limiting the generalizability of the reported findings. Finally, some methodological limitations should be addressed and/or some details be clarified.

In detail:

- The attrition rate of approximately 50% between baseline and 1-year follow-up acquisitions is relevant. Due to clinical heterogeneity between dropped-out and followed-up patients, it is hard to compare cross-sectional and longitudinal results for any of the reported MRI measure. This aspect should be discussed to explain the "lesser observed longitudinal changes" mentioned in the discussion, and results of cross-sectional baseline analysis of the 11 patients who underwent follow-up should be provided.
- The clinical phenotype of the included sample is unusually benign, for being ALS cases, as some patients have a disease duration longer than 10 years and the average ALSFRS-R slope is -0.4 points/month. This limits the generalizability of the authors' findings. Moreover, Table 1 should report standard deviations of all continuous variables, including age and disease duration.
- Was genetic assessment performed in the included patients? Some genetic alterations (e.g., SOD1 and C9orf72 mutations) have shown atypical MRI patterns, also in the spinal cord.
- Apparently, CSA was defined based on diffusion MRI data. This is not an anatomical sequence and, therefore, not suitable for volumetric assessment when it is used alone.
- The tract-based analysis is highly dependent on accurate segmentation, which was visually checked. How was the confidence value of 0.1 established?
- In the statistical analysis, a correction for multiple testing is mentioned. What kind of correction was used exactly?

Reviewer #2 (Remarks to the Author):

There are few studies on the spinal cord with longitudinal examination and correlation with functional status. This study was aimed to confirm and extend previous findings while addressing of the limitations of prior longitudinal study of spinal cord in ALS. Moreover, the study was aimed to identify

the spinal levels most affected in the disease. These aspects are most important to underline the use of DTI studies as tool in the clinical management of ALS and to improve the knowledge of the disease. The study is well-done and well-written. The originality of the work compared to the other studies is the analysis of individual fiber tracts in spinal cord of ALS patients.

However, there are some concerns to be addressed. Taking into account that the greatest significance of differences is noted in the C2-C3 and one of the muscle innervated by this area is the diaphragm, the study should also be completed with an analysis of possible correlation between FA or CSA with respiratory function. Furthermore, it should be interesting to evaluate a possible difference in the ALS group between the upper limb onset with the other type of onset to understand if the results are influenced by the first subgroup.

As minor concern, please include this study: Patzig M, et al. Measurement of structural integrity of the spinal cord in patients with amyotrophic lateral sclerosis using diffusion tensor magnetic resonance imaging. PLoS One 2019. Because they also found that FA was significantly reduced in ALS patients in anterolateral ROIs and the whole cross section at the C2-C4 level compared to healthy controls.

Reviewer #3 (Remarks to the Author):

In this study DTI metrics were assessed along the cervical spinal cord in 20 patients with ALS and 20 healthy controls. In addition longitudinal data were acquired from 10/11 ALS and 14/13 control participants, at 6/12-month follow-ups respectively. They performed a tract-specific analysis using a semi-automated method to segment the spinal cord. This allows to evaluate the continuous variation of DTI metrics along the spinal levels.

The study is well performed, well written and provides original and novel data. The strength of the paper is the that the used approach may result in more accurate intergroup comparison.

As in many ALS studies (and as stated as limitation) the sample size is quite low.

Overall I only have some minor comments:

- Figure 4. Why do you use 25th and 75th percentiles, and the range of FA / CSA. For a better illustration of group differences and the statistical significance, please use 95%CI.

- I recommend to avoid terms like "highly significant". The result is significant or not. There is no high or low significance. This term implies a higher relevance when the p-value is lower. However, this is - from a statistical point of view - not the case.

- Given the different cerebral DTI patterns between bulbar and limb onset, I wonder if there is a difference between both onset types in the cervical spinal cord?

You excluded the bulbar items from the ALSFRS-R (methods line 462). Did you exclude the bulbar onset patients from the correlation analyses. If yes, please mention this in the methods. If not, please clarify.

- You might discuss the involvement of sensory pathways in more detail.

Response to Reviews

We thank the Editor and the Reviewers for their valuable and insightful comments, which helped us improve the manuscript. We have completed a substantial revision of the manuscript, and added content to the results and discussion sections, based on the comments we received (major additions to the paper are highlighted in blue text). Below we provide point-by-point answers to the Reviewers:

Editor's comments

General comment: *Your manuscript entitled "Tract-specific analysis improves sensitivity of spinal cord diffusion MRI to cross-sectional and longitudinal changes in Amyotrophic Lateral Sclerosis" has now been seen by 3 referees, whose comments are appended below. You will see from their comments copied below that while they find your work of considerable potential interest, they have raised quite substantial concerns that must be addressed.*

At this stage, reviewer #1 has raised important concerns about the generalization of the approach based on the atypical clinical features of the participants and the small n number. Thus, we ask that you please address this concern with additional data or further characterization of the subject pool. Reviewer #2, also requests the addition of experiments to determine the physiological relevance of the changes observed, which are important experiments to include.

Our response: [Please also refer to our replies to Reviewer 1's and Reviewer 2's comments]

We had 20 ALS and 20 control subjects at baseline, and 11 ALS and 13 control subjects at one year follow-up. Despite the relatively small sample size, we believe that factors such as the quality of the data (high spatial and angular resolution, collected using advanced protocols and corrected for distortions using state-of-the-art techniques), and the robustness of the methods we used (filtering out covariates such as age and scanner type) led us to important results. We conducted an 'along-the-tract' analysis (that is, analysis of the continuous variations of dMRI metrics along the tract, instead of analyzing individual spinal level means, as done in all previous studies) in this study, and as far as we know, **our work is the first study reporting the continuous variation in change of diffusion metrics along the cord**, extracted using fiber tractography.

We agree that the sample size, especially in the longitudinal analysis, is small. We also acknowledge that drop-out due to disease progression is an inherent problem in longitudinal studies in ALS. We have described this as a limitation of our study in the 'Limitations' section of the 'Discussion'. Our cohort is atypical in the slow progression of clinical functional loss. With regard to other clinical characteristics, such as age (mean 57.5 years), site of onset (majority limb-onset), and gender (slight majority male), it is not atypical of the ALS population [1,3,4,5]. **We feel that the sole atypical feature, slow progression, in fact highlights the value of our findings.**

We have provided a detailed characterization of the cohort in Table 1. We also performed further characterization and subgroup analyses based on the site of onset, as suggested by Reviewers 2 and 3.

Additionally, we have also conducted the additional experiments which Reviewer 2 had suggested, to determine the physiological relevance of the changes observed (correlation between FA and CSA at C2-C3 with respiratory function).

Reviewer #1

General comment: *This manuscript reports the results of a cervical cord diffusion MRI study performed on 20 ALS patients, 11 of whom also underwent follow-up for up to one year from baseline. Both baseline cross-sectional and longitudinal analyses of whole-cord and tract-specific diffusion MRI metrics, as well as cervical cord cross-sectional area were assessed. Patients were compared with 20 age- and sex-matched healthy controls (13 of which underwent follow-up at 1 year). A semi-automated procedure for the segmentation of cord tracts was used, demonstrating increased sensitivity of mean FA in distinguishing ALS patients from healthy subjects in the cross-sectional analysis, particularly when only the descending tracts were considered. The 1-year longitudinal analysis showed significant progression of RD and MD in ALS patients, whereas these values remained stable in healthy controls. Significant correlations between FA and measures of clinical impairment were also found in the cross-sectional analysis.*

The main novelty of this study is the application of a semi-automated method for the definition of specific tracts within the cervical cord using diffusion MRI data, in order to identify sensitive biomarkers of ALS progression that do not depend on the delineation of ROIs by an operator. Moreover, only a few longitudinal studies assessed cervical cord alterations in ALS. However, the results are not ground-breaking per se (as they are in line with previous reports) and the sample size is small, therefore reducing the significance and complicating the interpretation of the results, particularly when cross-sectional and longitudinal results are compared. The clinical features of the included sample are also atypical, limiting the generalizability of the reported findings. Finally, some methodological limitations should be addressed and/or some details be clarified.

Our response: We thank the Reviewer for acknowledging the novelty of our study. We conducted additional experiments, mainly to address the concern about the atypical / heterogeneous clinical features of the cohort, and have extended the discussion section of the paper, as detailed below.

Our results are in line with previous reports, however our work substantially added to prior findings, making it unique. As far as we know, our **work is the first study reporting the continuous variation in change of diffusion metrics along the cord**, extracted using fiber tractography, instead of analyzing the spinal level means, as done in all previous studies. The proposed tractography-based approach facilitates between-subject statistical analysis and is tailored to the specific anatomy of each participant (see

reference 14), potentially resulting in more accurate inter-group comparison. In addition, we conducted a comprehensive analysis of ALL the fiber tracts along the cord (and sub-groups such as the full white matter or only descending tracts), and reported results from all the additional tracts (e.g. spinal lemniscus) in which we found significant changes, compared to selective analyses conducted in previous studies. We also reported the correlations of diffusion metrics with the ALSFRS-R scores for the cross-sectional data.

Our cohort is atypical in the slow progression of clinical functional loss. With regard to other clinical characteristics, such as age (mean 57.5 years), site of onset (majority limb-onset), and gender (slight majority male), it is not atypical of the ALS population. We feel that the sole atypical feature, slow progression, in fact highlights the value of our findings: **Our method is powerful, with increased sensitivity through tract-specific analysis, despite the early stage and slow progression of the patients cohort.** Specifically, we observed a statistically significant longitudinal change in RD and MD in numerous analyses of ALS subjects only, despite a small sample size of largely slow progressors. This suggests that this technique will be highly sensitive to change in a more typical cohort, and possibly over a shorter period of observation. We have expanded on this in the discussion section to address the Reviewer's question.

We agree that there exists heterogeneity in the clinical features of the disease. However, our experiments are designed in such a way to identify the changes that are 'common' to a given group, and not the changes which are specific to any of its subgroups. In other words, we are reporting only the 'generalizable' findings. The utility of the method could potentially be increased if we analyzed subgroups within the ALS cohort. Though we conducted such analyses, we have decided not to include those results in the paper as the number of subjects in each subgroup is very small. However, we have added the conclusions which we could make from subgroup analyses, based on the site of onset and on disease severity, to the discussion section (see details below).

We also agree that the number of participants (11) from which we could collect longitudinal data is low, due to dropout, **which is an inherent problem in longitudinal studies in ALS** (we describe this as a limitation of our study in the 'Limitations' section of the 'Discussion'). The strength of our contribution is the quality of our data and the rigor of our analysis methods. This study serves as a proof-of-principle and can be used to design a scalable protocol that can be applied in clinical and clinical research settings.

Comment-1: - *The attrition rate of approximately 50% between baseline and 1-year follow-up acquisitions is relevant. Due to clinical heterogeneity between dropped-out and followed-up patients, it is hard to compare cross-sectional and longitudinal results for any of the reported MRI measure. This aspect should be discussed to explain the "lesser observed longitudinal changes" mentioned in the discussion, and results of cross-sectional baseline analysis of the 11 patients who underwent follow-up should be provided.*

Our response: We agree with the Reviewer that it is difficult to compare cross-sectional differences between ALS and control cohorts, with longitudinal changes within the ALS cohort. We have therefore removed this comment from the paper. We also conducted a cross-sectional baseline analysis of the 11 patients who underwent follow-up. We have added a description of this analysis to the results section, and added the data as a new figure in the supplementary material (Supplementary Fig. 5).

Comment-2: - *The clinical phenotype of the included sample is unusually benign, for being ALS cases, as some patients have a disease duration longer than 10 years and the average ALSFRS-R slope is -0.4 points/month. This limits the generalizability of the authors' findings. Moreover, Table 1 should report standard deviations of all continuous variables, including age and disease duration.*

Our response: This concern is addressed above (please refer to our answer to Reviewer 1's general comment). We have modified Table 1 to report standard deviations of all continuous variables, including age and disease duration, as requested.

Comment-3: - *Was genetic assessment performed in the included patients? Some genetic alterations (e.g., SOD1 and C9orf72 mutations) have shown atypical MRI patterns, also in the spinal cord.*

Our response: Because the likelihood of obtaining a meaningful number of subjects with a particular genetic etiology was negligible, genetic testing was not part of the study protocol.

Comment-4: - *Apparently, CSA was defined based on diffusion MRI data. This is not an anatomical sequence and, therefore, not suitable for volumetric assessment when it is used alone.*

Our response: The spinal cord toolbox (SCT) provides an option to calculate the cross-sectional area (CSA) from the diffusion MRI data. It preprocesses the diffusion MRI data for motion correction, and calculates the mean of the diffusion-weighted images in which the spinal cord is bright and easily identifiable (see the included images below) as in T1 images (though other tissues are not clearly visible). The spinal cord is subsequently segmented from the mean diffusion-weighted image using a deep learning algorithm, which provides high quality segmentation of the cord (see images from two different subjects below). The CSA is calculated based on the voxel size and the number of voxels in this segmentation. As these are the fine details described in the SCT documentation, we have not explained the segmentation procedure in the paper. However we have cited the appropriate references [32,33].

We also confirmed with the developers of SCT (as indicated in the Acknowledgement section) that the CSA calculation was done appropriately in our study. We also would like to emphasize that the diffusion MRI data was collected with an in-plane (transverse) spatial resolution of $1.12 \times 1.12 \text{ mm}^2$, despite a greater slice thickness (3.3 mm) which does not affect the CSA computation. We therefore believe that our data is adequate to accurately estimate the CSA.

Comment-5: - The tract-based analysis is highly dependent on accurate segmentation, which was visually checked. How was the confidence value of 0.1 established?

Our response: We established this threshold by manual inspection of the segmentations, as mentioned in the paper. However, the results and findings described in the paper are not sensitive to this threshold (as long as we use the same threshold for all the subjects; we have verified this with additional experiments), as we are not reporting the absolute volumes of individual fiber tracts, but rather the diffusion features in these tracts. We have clarified this point in the paper. The accuracy of the segmentation was actually an issue for the rubrospinal tract due to its small size and overlap with the corticospinal tract. As such, we have excluded the data from the rubrospinal tract.

Comment-6: - In the statistical analysis, a correction for multiple testing is mentioned. What kind of correction was used exactly?

Our response: The corrected p-value is calculated using a wild bootstrap method, as below.

$$p(s_m) = G^{-1} \sum_{g=1}^G 1(S_{n,\max}^{(g)} \geq S_n(s_m))$$

where,
 $p(s_m)$

is the corrected p-value at grid point (tract location) s_m ,

G

is the number of bootstraps, $G = 1000$ in our case,

$S_{n,\max}^{(g)}$

is the maximum of the test statistic at all grid points, for the bootstrap sample g , and

$S_n(s_m)$

is the local test statistic at grid point (tract location s_m).

The calculation of p-value as above corrects the p-value based on the number of tract locations tested. We added the following brief description of the method to the manuscript and have cited the appropriate reference [13]; due to space limitations, we did not feel that most readers would find value in seeing the above detailed description of the method.

“The p-value calculation is done with a wild bootstrap method, by checking whether the local test statistic at each tract location is greater than the maximum of the test statistics at all grid points along the tract (that is the number of multiple tests), for all the bootstrap samples, which corrects for multiple testing along the cord.¹³”

Reviewer #2

General comment: *There are few studies on the spinal cord with longitudinal examination and correlation with functional status. This study was aimed to confirm and extend previous findings while addressing of the limitations of prior longitudinal study of spinal cord in ALS. Moreover, the study was aimed to identify the spinal levels most affected in the disease. These aspects are most important to underline the use of DTI studies as tool in the clinical management of ALS and to improve the knowledge of the disease.*

The study is well-done and well-written. The originality of the work compared to the other studies is the analysis of individual fiber tracts in spinal cord of ALS patients.

However, there are some concerns to be addressed.

Our response: Our sincere thanks to the Reviewer for acknowledging the originality and importance of our work to aid the clinical use of DTI for diagnosis and disease management. We addressed all the expressed concerns, as detailed below.

Comment-1: *- Taking into account that the greatest significance of differences is noted in the C2-C3 and one of the muscle innervated by this area is the diaphragm, the study should also be completed with an analysis of possible correlation between FA or CSA with respiratory function.*

Our response: We performed an analysis of the possible correlation between FA and CSA at C2-C3, with respiratory function (measured with questions 10-12 of ALSFRS-R), and did not find any significant correlations. In fact, 18 out of the 20 ALS participants had a score of 12 out of 12 for the respiratory function. One participant had a score of 11 and another had 10. This ruled out the chance for any correlations. We also studied the data at 1-year follow-up, where the respiratory function score reduced in a few participants, but again we did not find any correlation either with FA or with CSA. We added this inference to the paper.

Comment-2: - Furthermore, it should be interesting to evaluate a possible difference in the ALS group between the upper limb onset with the other type of onset to understand if the results are influenced by the first subgroup.

Our response: We thank the Reviewer for this comment, which helped us perform additional and important experiments. We addressed this comment together with another similar comment (comment-3) of Reviewer-3, in which the Reviewer asked to study the difference between bulbar vs. other onset patients. As such, we conducted the following new analyses:

- a) Group difference in FA along the lateral CST **between upper limb and other type of onset** (bulbar and lower limb) patients. We noted that upper limb onset patients have lower mean FA compared to other onset patients, but this difference did not reach a statistically significant level (please refer to newly added figure, Supplementary Fig. 4).
- b) Group difference in FA along the lateral CST **between bulbar and other type of onset** (lower and upper limb) patients. We noted that (upper and lower) limb onset patients have lower mean FA compared to bulbar onset patients, but this difference did not reach a statistically significant level.
- c) Group difference in FA along the lateral CST **between upper and lower limb onset patients**. We noted that upper limb onset patients have lower mean FA compared to lower limb onset patients, but again this difference did not reach a statistically significant level.

In other words:

(meanFA(upper limb onset)=0.45) < (meanFA(lower limb onset)=0.47) < (meanFA(bulbar onset)=0.50) < (meanFA(control)=0.53)

We did similar analyses for the CSA of the whole cord and found the following:

(meanCSA(upper limb onset)=56.7 mm²) < (meanCSA(lower limb onset)=58.4 mm²) < (meanCSA(bulbar onset)=65.5 mm²) < meanCSA(control)=69.0 mm²).

We have added these results to the Results section, and a new figure in the supplementary material (Supplementary Fig. 4). The description of the procedure is added to the Methods section and a brief discussion is also added to the Discussion section.

Comment-3: - As minor concern, please include this study: Patzig M, et al. Measurement of structural integrity of the spinal cord in patients with amyotrophic lateral sclerosis using diffusion tensor magnetic resonance imaging. PLoS One 2019. Because they also found that FA was significantly reduced in ALS patients in anterolateral ROIs and the whole cross section at the C2-C4 level compared to healthy controls.

Our response: We thank the Reviewer for this suggestion to refer and cite this very recent cross-sectional study in ALS using spinal cord diffusion MRI. We have now referenced and cited this work. Our work is very

different from this study as our approach is a 'tract-specific' and 'along-the-tract' analysis, and as we analyze both cross-sectional and longitudinal data. We also analyze the diffusivity measures (RD, MD, and AD) and cross-sectional area (CSA) in contrast to the analysis of only FA in the cited work.

Reviewer #3

General comment: *In this study DTI metrics were assessed along the cervical spinal cord in 20 patients with ALS and 20 healthy controls. In addition longitudinal data were acquired from 10/11 ALS and 14/13 control participants, at 6/12-month follow-ups respectively. They performed a tract-specific analysis using a semi-automated method to segment the spinal cord. This allows to evaluate the continuous variation of DTI metrics along the spinal levels.*

The study is well performed, well written and provides original and novel data. The strength of the paper is the that the used approach may result in more accurate intergroup comparison. As in many ALS studies (and as stated as limitation) the sample size is quite low.

Overall I only have some minor comments:

Comment-1: *- Figure 4. Why do you use 25th and 75th percentiles, and the range of FA / CSA. For a better illustration of group differences and the statistical significance, please use 95%CI.*

Our response: We thank the Reviewer for this comment to improve our figure with 95% CI. We have modified Figure 4 accordingly: the box plots with 25th and 75th percentiles, and the range of FA / CSA are replaced with 95% CI. We hope this illustrates the group difference better.

Comment-2: *- I recommend to avoid terms like "highly significant". The result is significant or not. There is no high or low significance. This term implies a higher relevance when the p-value is lower. However, this is - from a statistical point of view - not the case.*

Our response: We have removed all these terms and replaced with "significant".

Comment-3: *- Given the different cerebral DTI patterns between bulbar and limb onset, I wonder if there is a difference between both onset types in the cervical spinal cord? You excluded the bulbar items from the ALSFRS-R (methods line 462). Did you exclude the bulbar onset patients from the correlation analyses. If yes, please mention this in the methods. If not, please clarify.*

Our response: We thank the Reviewer for this question, which led to additional experiments to confirm our prior hypothesis that there was no significant difference between bulbar and limb onset participants' spinal cord data. Because of this, we have not excluded the bulbar onset participants from the correlation analysis,

though we excluded the bulbar functional component scores from the ALSFRS-R, as there are no corticobulbar projections in the spinal cord. However, we noted some differences between these groups, as detailed below.

We address this comment together with another similar comment (comment-2) of Reviewer-2, in which the reviewer asked to study the difference between upper limb vs. other onset patients. As such, we conducted the following analyses:

- a) Group difference in FA along the lateral CST **between upper limb and other type of onset** (bulbar and lower limb) patients. We noted that upper limb onset patients have lower mean FA compared to other onset patients, but this difference did not reach a statistically significant level (please refer to newly added figure, Supplementary Fig. 4).
- b) Group difference in FA along the lateral CST **between bulbar and other type of onset** (lower and upper limb) patients. We noted that (upper and lower) limb onset patients have lower mean FA compared to bulbar onset patients, but this difference did not reach a statistically significant level.
- c) Group difference in FA along the lateral CST **between upper and lower limb onset patients**. We noted that upper limb onset patients have lower mean FA compared to lower limb onset patients, but again this difference did not reach a statistically significant level.

In other words,

$(\text{meanFA}(\text{upper limb onset})=0.45) < (\text{meanFA}(\text{lower limb onset})=0.47) < (\text{meanFA}(\text{bulbar onset})=0.50) < (\text{meanFA}(\text{control})=0.53)$

We did similar analyses for the CSA of the whole cord and found the following:

$(\text{meanCSA}(\text{upper limb onset})=56.7 \text{ mm}^2) < (\text{meanCSA}(\text{lower limb onset})=58.4 \text{ mm}^2) < (\text{meanCSA}(\text{bulbar onset})=65.5 \text{ mm}^2) < \text{meanCSA}(\text{control})=69.0 \text{ mm}^2$.

We have added these results to the Results section and a new figure in the supplementary material (Supplementary Fig. 4). The description of the procedure is added to the Methods section and a brief discussion is also added to the Discussion section.

Comment-4: - *You might discuss the involvement of sensory pathways in more detail.*

Our response: Further discussion on the involvement of sensory pathways was added to the Discussion section (in the subsection "Insights into pathological changes in ALS"), as requested.

---- END ----

REVIEWERS' COMMENTS:

Reviewer #1 (Remarks to the Author):

The authors addressed comments properly

Reviewer #2 (Remarks to the Author):

No further comments.

The author well-addressed all my concerns. In this version the manuscript is suitable for publication

Reviewer #3 (Remarks to the Author):

In the revised version all my concerns were addressed.